# Neglected Occupational Risk Factors—A Contributor to Diagnostic Delays in Lung Cancer

**DOI:** 10.3390/healthcare14010106

**Published:** 2026-01-01

**Authors:** Cristina Mandanach, Andreea Maftei, Ocxana Maria Țocan, Claudia Lucia Toma, Marina Ruxandra Oțelea

**Affiliations:** 1Doctoral School, “Carol Davila” University of Medicine and Pharmacy, 37 Dionisie Lupu Street, Sector 2, 020021 Bucharest, Romania; cristina.paraschiv@drd.umfcd.ro (C.M.); andreea.mutu@drd.umfcd.ro (A.M.); ocxana-maria.tocan@drd.umfcd.ro (O.M.Ț.); 2Clinical Department 5, “Carol Davila” University of Medicine and Pharmacy, 37 Dionisie Lupu Street, Sector 2, 020021 Bucharest, Romania; 3Pulmonology Department I, “Carol Davila” University of Medicine and Pharmacy, 37 Dionisie Lupu Street, Sector 2, 020021 Bucharest, Romania; claudia.toma@umfcd.ro; 4Pulmonology Department IV, “Marius Nasta” Institute of Pneumology, 90 Viilor Street, Sector 5, 050159 Bucharest, Romania; 5Clinic of Occupational Medicine, Colentina Clinical Hospital, 19–21 Stefan cel Mare Street, Sector 2, 020125 Bucharest, Romania

**Keywords:** occupational carcinogens, occupational exposure, lung cancer, total interval time to diagnosis

## Abstract

**Highlights:**

**What are the main findings?**
Due to the long period between the first exposure and the first signs of lung cancer, occupational risk is under evaluated.Patients with occupational risk of lung cancers have longer time to diagnosis.

**What are the implications of the main findings?**
Occupational hazards should be included in the risk assessment in order to reduce the time to diagnosis of patients with lung cancer.Follow-up of the workers exposed to lung carcinogens in their workplace could lead to an earlier detection of lung cancer.

**Abstract:**

Introduction: For lung cancer, the total interval time to diagnosis (TITD) is very important. If not detected by the screening program, the actual guidelines emphasize the need for a short delay to assure the initiation of treatment before 2 months from the initial symptoms. In order to shorten TITD, the individual risk has to be properly assessed by the primary physician. Objective: The assessment of the influence of the occupational exposure on the diagnostic delay—from the onset of symptoms to confirmed diagnosis—in a population of patients with lung cancer. Material and methods: A total of 110 cases were recruited and were divided into two groups based on the individual assessment by an occupational physician. Results: There were 38 cases (34.55%) at high risk according to their occupational exposure and 72 controls. On average, the TITD was 3.41 +/−5.12 months. The TITD was significantly longer in the high-risk group (*p* = 0.03). A larger proportion cases had longer TITD: 55.17% of cases vs. 44.83% of controls (*p* = 0.006). In a multivariate analysis including covariates’ age, sex, level of education, health literacy, number of packs-years, family history of cancer, and previous lung diseases associated with a high risk of lung cancer, the highest risk derived from the previous occupational exposure was the only variable statistically associated with TITD (OR = 10.57, 2.06–54.34, *p* = 0.01). Discussion: Awareness about the occupational risk in workers who are or have been exposed and in health providers could reduce the total interval time to diagnosis.

## 1. Introduction

Lung cancer (LC) remains one of the most important issues in public health, as mortality is the greatest of all malignant diseases, even in economically developed countries. The 5-year relative survival is 28.1% [1] and the mortality rate of lung cancer in the European countries was 77.2 per 100,000 men and 32.8 per 100,000 women [2]. These general European data are reflected also in the Romanian statistics [3], with LC occupying the first rank in mortality of all cancers. Currently, low dose computed tomography (LDCT) scan is the only lung cancer screening method recommended by major medical bodies such as the U.S. Preventive Services Task Force (USPSTF) and the European Society of Radiology, specifically for high-risk populations, primarily based on smoking history [4]. In some countries in which post-exposure medical surveillance is in place for certain occupational hazards (e.g., previous occupational exposure to asbestos) [5], the number of occupational LC is relatively high. Limited data are available and neither USPT nor European medical guidelines are conclusive for other exposures [6]. Screening and early diagnosis are strongly justified, as the 5-year survival rate in non-small cell lung cancer (NSCLC) drops from over 90% in stage IA to less than 10% in stage IV [7]. As with other cancers where screening programs are implemented, in the absence of such procedures, early diagnosis relies on minimizing diagnostic delays. In European countries, according to the different guidelines in place, this period should be between 8 and 62 days from the initial presentation [8] and treatment should be initiated at maximum 2 months [9].

The interval time to diagnosis and treatment is an important factor of survival, as the time for doubling of the tumor size varies between 70 and 260 days [10]. The non-adenomatous forms have a faster rate of growth, but the distinction is not always obvious before the histology result is available. Therefore, as a matter of precaution, the shorter TITD is considered better.

The pathway from first symptoms to diagnosis and treatment in LC is influenced by a multitude of variables. It includes both the patient interval (time from symptoms to first visit) and the doctors’ interval (time from first visit to diagnosis) [11]. Both intervals are influenced by many actors of the medical and social system. Part of this delay is related to the access to the medical care, and to the self-perception of the risk and of the personal symptoms management [12]. The occupational risk is generally underestimated by the workers [13]. Openness and emotional stability seem to be associated with a reduction in risk perception, while conscientiousness and friendliness with an increase [14].

One of the most frequent factors of delay is a low index of suspicion in the primary care services [8] and occupational exposure is frequently not included in the risk assessment. Although highly under-diagnosed, we know from epidemiology research that occupational LCs represent at least 8–15% of the LCs [15,16,17] and covers all types of histological forms. Some occupations, such as agriculture (exposure to pesticide), mining and quarrying (exposure to silica), welders (exposure to vapors, fumes, metals, and dust), metal industries (exposure to metal alloys), construction, shipyards, insulators (exposure to asbestos), and heavy industry (exposure to metals and polycyclic aromatic hydrocarbons) are “classical” examples of high prevalence. To assess awareness of occupational exposure, we analyzed whether this variable influenced the diagnostic delay—from the onset of symptoms to confirmed diagnosis—in a population of patients with lung cancer.

## 2. Materials and Methods

Data presented in this analysis are part of a larger study concerning the evaluation of occupational exposure in patients with cancer and refer only to the lung cancer data. For this part of the study, recruitment of a convenience sample of patients diagnosed with LC was performed in the Marius Nasta Institute of Pulmonology and in the Colentina Clinical Hospital, after the protocol was approved by the Ethical Committees of each of these centers (Colentina Clinical Hospital no. 20/23 November 2022; Marius Nasta Institute of Pneumophthisiology no. 23941/25 October 2023). Although a convenience sample is not representative for the whole patients’ population, in order to reduce possible biases, we included patients from the largest hospital in Bucharest fully dedicated to lung diseases (Marius Nasta Institute of Pulmonology), and a large general, multidisciplinary hospital. Both hospitals had pulmonology and oncology departments and the patients diagnosed with cancer were also receiving the adjuvant oncological treatment and followed periodically. Criteria for inclusion were the following: age > 18 years, a histological confirmation of LC, and signed informed consent. There were no exclusion criteria based on cellularity. Patients who declared having difficult access to medical services were excluded. Cases were recruited randomly from patients who were scheduled for the follow-up consultation and/or ambulatory treatment on consecutive Thursdays.

A structured interview was conducted by an occupational medicine physician. The interview included demographic data (age, gender, level of instruction, occupation), family history of cancer, exposure data (smoking, occupation, occupational hazards). Data related to smoking history included the age of smoking initiation, total duration of smoking (in years), and the average number of cigarettes smoked per day. Based on the number of years of smoking and the average number of cigarettes/days, the number of pack years was calculated. Recent publications have enlarged the indication for screening from the initial threshold of 20 pack-year to only longer than 20 years of smoking, even with fewer cigarettes smoked/day [18]. Therefore, we have included in the group of high risk related to smoking both elements either more than 20 pack years or more than 20 years of smoking.

Data related to diagnostic and comorbidities associated with a high risk of lung cancer, such as chronic obstructive pulmonary disease (COPD) or lung fibrosis, were extracted from the patients’ medical files. All pulmonary diseases were certified by a pulmonologist.

The short version, with 16 items of the European Health Literacy Survey Questionnaire, validated in the Romanian population, was applied [19] to assess the patients for the four core competencies of health literacy (find, understand, appraise, and apply health information) in three domains (healthcare, disease prevention, health promotion). According to the guidelines for interpretation, the answers “easy” and very easy” were scored with 1 and the ones “very difficult” and “difficult” with 0. The total score (HLS-EU-Q16) could be 16 and the health literacy is considered inadequate (0–8), problematic (9–12), and adequate (12–16) [20]. For the regression model with total interval time to diagnosis (TITD) as outcome, the raw data of all answers was used as covariable.

In addition to this overall score, four specific questions were considered relevant for the outcome and were separately analyzed. These are the following:

“On a scale from very easy to very difficult, how easy would you say it is to:(a)Find out where to get professional help when you are ill?(b)Understand what your doctor says to you?(c)Use information the doctor gives you to make decisions about your illness?(d)Understand health warnings about behavior such as smoking, low physical activity and drinking too much?”

Questions (a) and (d) were chosen for their possible influence on the time from symptoms to first visit. The questions (b) and (c) were mainly related to the time from first visit to diagnosis. The maximum score for each of these questions is for comparison, inadequate score was considered if the answer was “very difficult” and “difficult”, and adequate if “easy” and very easy”.

The outcome measured for this study was the time (in months) from first symptoms to diagnosis, further referred as total interval time to diagnosis (TITD). As there is no data available at national level, we first calculated the average TITD. Patients within the upper half of the TITD, considered as having a long time, were compared with those with shorter than the average TIID.

After the data collection was completed, two independent occupational physicians classified the patients according to their declared exposure to LC occupational hazards. Classification was blinded for both evaluators. Each expert classified the exposure based on hazards (namely the ones included by IARC in the class I category), occupation, and occupational domain. The inter-rater agreement was calculated with MedCalc statistical software (https://www.medcalc.org/en/calc/kappa.php (Version 23.4.5; accessed 31 December 2025)) [21]. When disagreement was met, a consensus was reached using as references for occupations/work process listed in the existing international data bases alte or published in the scientific literature [22,23,24].

The ones with declared exposure or with evidence of risk related to their occupation were included in the cases group and all others in the control group.

There are currently different approaches for screening in occupational medicine; most of them recognize 10 years as minimum exposure [6,25,26] except for very high exposure that is also mentioned in some national recommendations [26]. However, in a large cohort of highly exposed workers, 5 years of exposure was not sufficient to support the implementation of the low dose CT screening [27]. The duration of exposure time is a significant element in calculating the lifetime cumulative exposure. Therefore, we have chosen the 10 years of exposure to be a better estimate of the high-risk category.

The latency for solid cancers is generally long, estimated to be more than 8–10 years [28,29]. An exposure time of 10 years would definitely include this latency and therefore will be a better indicator of high risk, as used in the screening recommendations described above. Even more, in the analysis of residents with LC (1042 exposed and 2364 controls, in Stockholm area), the sensitivity analysis based on the latency of exposure did not influence the overall results [30], an argument in favor to our classification.

Numerical variables were first assessed for normality. Statistical difference for the qualitative variables was computed using chi square test; for the numerical ones, Anova or Mann–Whitney test was used, according to the normality of the distribution. Binary regression was calculated for the association between high occupational risk and delayed TITD. Further on, we verified the influence of other variables on this correlation. Besides the basic demographic data (age, gender), we included other well-defined risk factors for LC (number of packs-years, pulmonary diseases associated with LC, and family history of cancer). The score of health literacy was used in order to exclude a possible bias derived from possible delays generated by the patient lack of understanding or risk factors or of doctors’ recommendations. Before proceeding to the multivariate regression, collinearity of the variables has been checked. The software for the statistical analysis was the StatPlus:mac, macOS Version, v8. provided by AnalystSoft Inc., Brandon, FL, USA.

## 3. Results

For an estimated one third of the population with a high risk of occupational exposure, the study required a sample of 107 patients with a level of confidence of 95% [31]. Therefore, 110 patients were recruited. Based on the histological classification, 44.94% were adenocarcinoma, 33.71% squamous cell carcinoma, 15.73% small cell lung carcinoma, and 5.62% other forms. Among these patients, 38 (34.55%) were at high risk according to their occupational exposure and 72 (65.45%) represented the controls. The inter-rater agreement between the two evaluators of the occupational risk was very high (kappa = 0.92848, standard error = 0.03512 and 95%CI: 0.85965–0.99731). For the four cases that were differently classified, a consensus was reached by discussing the initial opinions and confronting them with the most relevant literature references for that particular patient.

The average age was 63.95 years (SD = 9.50). Most of them were men (58.75%). Smoking was frequent (80.00%) in the total sample, and most were heavy smokers (87.50%). The majority of patients smoked for more than 20 years. Only four patients (two with occupational exposure to carcinogenic hazards and two without) smoked less than 20 years.

The average HLS-EU-Q16 score was 13.24 (SD = 2.97). A total of 8 patients had an inadequate score, 29 had a problematic one, and 70 (63.63%) had an adequate health literacy level. Inadequate and problematic scores were detected in 13 (34.21%) cases and in 23 (31.94%) controls (χ^2^ =0.61, *p* = 0.73). The distribution of level of health literacy is presented in Figure 1.

No correlation between HLS-EU-16 and age at diagnosis and TITD was found. There was a significant, direct correlation between HLS_EU_16 score and the level of education (R = 0.64, *p* < 0.000). There was a borderline inverse correlation between the HLS-EU-16 score and the number of pack years smoked (R = −0.18, *p* = 0.06).

There were 6 patients (5.5%) declaring to find it fairly difficult to locate professional help for illness, 19 (17.27%) with difficulties in understanding of what the doctor said, 18 (16.36%) in using the doctors’ information to make decisions, and 12 (10.9%) in understanding the risk factors for health, including smoking. The average scores were inside the adequate level range for all these four questions (Table 1).

The average TITD was 3.41 months. The majority of patients were diagnosed in less than 4 months, 30 in less than 1 month and 60 between 1 and 4 months. Fifteen patients were diagnosed between more than 4 months and 12 months and five in more than 12 months. The distribution according to the TITD is presented in the Appendix A.

Forty-eight patients (43.64% of the total sample) were exposed to at least one occupational carcinogen during their professional activity. Among this, 38 were exposed for longer than 10 years and were considered a high-risk group. These high-risk group patients represented the cases and all the others (72 patients), the controls.

In the cases group, the most frequent exposures were diesel exhaust (11 patients), silica dust (7 patients), polycyclic aromatic hydrocarbons (6 patients), and welding fumes (5 patients). A history of arsenic or arsenic compounds exposure was found in three patients. For each of the following hazards (asbestos, bis-chloromethyl ether, iron oxide), two cases were reported.

The characteristics of the study population are presented in Table 1. There were no statistically significant differences in age, gender, level of education, smoking, or health literacy among cases and controls.

A family history of cancer was present in 27 patients (24.55%) of the total sample. There were 11 patients (10% of the total sample) with a history of pulmonary cancer in the family. Among these, 7 patients were smokers, and 4 non-smokers. All the non-smokers were in the control group.

Of the 25 non-smokers in the whole sample, 8 were cases with known history of exposure to carcinogens and none of them had a family history of cancer. On the contrary, in the control group, 4 out of 17 non-smoker patients had a family history of cancer.

Cases had significantly more pulmonary diseases considered to be a risk factor for LC (χ^2^ = 5.65, *p* = 0.02). The most frequent was COPD (16 patients). One patient had TB and one was previously diagnosed with pneumoconiosis. There was no statistically significant difference in TITD inside the cases group between those with COPD or lung fibrosis and those without (χ^2^ = 0.48, *p* = 0.48).

Cases had a higher risk of having longer than 3.5 months total interval time to diagnosis: OR = 3.30 (95%CI: 1.37–7.96), *p* = 0.007. There was no collinearity of the variables included in the analysis. In the multivariate model, the statistical difference was maintained (Table 2).

## 4. Discussion

### 4.1. Occupational Risk and the Total Diagnosis Interval Time

The main finding of this study is that patients with long time occupational exposure to evidence-based carcinogens do not benefit from a shorter interval to diagnosis.

Awareness of the relation between occupational hazards and initial symptoms among those who are or have been exposed is important to generate a prompt visit to a healthcare provider. From the public health perspective, these are the duties of the employer and of the medical surveillance provided during and after the exposure. The occupational medicine doctors and nurses are responsible for the health education of the workers in relation to the occupational hazards and for the initial screening. Overall, a better, personalized communication with the workers exposed to occupational risk should shorten the time from symptoms to first visit.

An important factor influencing the time from first visit to diagnosis is the latency—the time from the initial exposure in the workplace to the first symptoms or subclinical signs (e.g., CT scan, if performed). This time is very long in solid tumors and can reach, for example, more than 30 years after the first exposure to asbestos [32]. This is why the majority of the occupational cancers are diagnosed after the retirement age, when other healthcare providers could be the first access point in the medical system. These healthcare providers are, generally, much less aware of the occupational related risk.

The time from first visit to diagnosis can be improved from better communication and involvement of the occupational medicine specialist from the initial steps of the diagnosis, either by providing a post-exposure surveillance in occupational medicine clinics [33] or by including this specialty in the clinical team. The occupational risk might be identified from exposure registries that serve as base for the post-exposure occupational medicine surveillance. If exposure registries are not available, data from country reports on prevalence of LC could be used. For example, in Switzerland, elementary professions in men in elementary professions and women in intermediate professions were associated with the highest risk of developing LC [34]. This study counted in the category of elementary professions “workers in agriculture, industry and crafts with basic level of education” and in the intermediate category “administrative employees and service personnel and intermediate technical occupations which generally do not involve planning or supervisory powers and in the managerial categories”.

From our national data, we have identified a risk for earlier age diagnosis of LC in leather industry, trade services, extractive industry, and wood and furniture domains [35]. At least when such data are available, they should be better communicated to all medical providers and included in the clinical practice guidelines.

### 4.2. Smoking in the Context of Occupational Exposure and the Total Diagnosis Interval Time

Smoking is well known as risk factor for LC in the general population [36] and in the medical community. However, a recent statistic showed the increase in percentage of non-smokers with LC [37]. On one side, this is the consequence of the decline in number of smokers and, on the other side, from a better understanding of the carcinogenic effects of other factors. Previous studies have shown, indeed, some additive or even multiplicative interaction between occupational carcinogens and smoking [16,38,39], but there are also many studies in non-smokers confirming that the risk of LC after exposure to asbestos, silica, and diesel exhaust [17] is high.

In fact, there is extensive evidence that LC in non-smokers is linked to exposure to occupational carcinogens. Any previous exposure to occupational exposures gives an OR = 2.1 (95% CI 1.3–3.3) in never smokers [40]. A pooled analysis of 14 case control studies showed that for never smokers, ORs for small cell lung cancer were more than double in both men and women with high cumulative exposure to polycyclic aromatic hydrocarbons [41]. Men working in the construction and petroleum industries and women working as farmers and as bakers had also a significantly higher risk of LC, which remained significant after we controlled for tobacco smoking and opium consumption [15]. Non-smoking women employed as painters or rubber workers had statistically significant higher relative risk of 2 and 1.7, respectively [42]. In our sample, smoking did not influence the link between LC and occupational carcinogens. This result just underlines the need to proceed to a definitive diagnosis whenever symptoms suggestive for LC are present. The lack of statistical association does not exclude the biological relation and the additive or synergic effects of smoking with the occupational hazards, supported by extensive medical data [43], and do not minimize the role of smoking itself in the risk assessment.

### 4.3. Family History of Cancer in the Context of Occupational Exposure and the Total Diagnosis Interval Time

It is also of note that cases with occupational exposure had lower rates of family history of cancer. This was true in particular for the non-smoking cases. The lower rate of family history of cancer could be another contributor to the delay in diagnosis, influencing both time from symptoms to first visit and the time from first visit to diagnosis. In what concerns the time to first visit, a study on a population screened for LC showed that the perception of risk in persons with family history of LC was higher than the perceived risk for cancer-related symptoms (dyspnea, cough) or COPD [36]. As a consequence, we could expect that people with a family history of cancer will seek a medical consultation more rapidly after the first symptom.

The prevalence of a family history of LC was lower in our study compared to the 16.6% prevalence in the pooled data of nine European cohorts, with a total of 216,387 participants [44]. Most studies support a polygenic influence on the LC risk, although the validation of applying a score risk at individual level is still uncertain [45].

In what concerns the comparison between cases and controls, the family history was more prevalent in controls, although not statistically significant. In interpreting this difference, we cannot exclude a possible bias coming from similar occupations (or occupational exposures) in the family members as we did not record the occupation of the family members. In order to clarify if this is a bias or not, further investigation, including genetic markers, focused on this topic is needed.

### 4.4. Pulmonary Diseases in the Context of Occupational Exposure and the Total Diagnosis Interval Time

COPD and LC have many common elements. Common risk factors and pathogenic mechanisms such as oxidative stress, cellular aging, genetic predisposition, and epigenetic modifications are well characterized [46]. The risk of LC is very high in smokers with COPD, but remains significant even in non-smokers with COPD [47].

In non-smokers, the prevalence and risk factors to develop COPD have been recently reviewed [48]. The most frequent causes of COPD in non-smokers were biomass smoke exposure from cooking and heating, occupational exposures, and outdoor air pollution [48]. While these causes are relevant for COPD, they are also relevant for LC [49,50]. There is no clear relation between the airflow limitation and the risk for LC, but the extension of emphysema increases the risk [40,51]. This relation is even stronger when emphysema is associated with lung fibrosis [52]. By itself, the presence of the lung fibrosis denotes a high risk of LC [53]. Idiopathic lung fibrosis (IPF) is a proliferative disease with deregulated cytokine control of the cellular expansion, genetic and epigenetic modifications. Some of these pathogenic elements are also frequent in LC. The evolution of LC in a fibrotic context is more aggressive [54]. Even if the occupational fibrosis is, in general, less aggressive than IPF, their progression despite exposure cessation proves an ongoing proliferative activity.

We found a significantly higher number of persons with pulmonary diseases in cases. This is not a surprising result as all the occupational hazards to which these patients have been exposed could lead either to chronic bronchitis, to emphysema, or to lung fibrosis. Although not recognized as occupational, probably mainly by the interference with smoking as risk factor, they should have raised the awareness to obtain a quicker diagnosis.

### 4.5. Health Literacy in the Context of Occupational Exposure and the Total Diagnosis Interval Time

In this study, there was no major difference in the level of education or in the total score of a validated health literacy questionnaire compared to the Romanian population. Compared to the initial validation study of the HLS-EU-16 questionnaire, our study population had more frequently sufficient health literacy (63.63% in our study compared to 59.2% in the validation study) [19]. The problematic and inadequate levels were rather similar (26% versus 33.2% and 7% versus 7.5%, respectively) [19].

From the questions which might have influenced the time from symptoms to first visit, the one about understanding health warnings about behavior such as smoking, low physical activity, and drinking had a marginally statistical significance. The questions related to the time from first visit to diagnosis gave similar scores.

There are few studies about health literacy and time to diagnosis. A systematic review failed to give an evidence-based conclusion, because very few studies (only three) retrieved from databases which specifically mentioned the time to diagnosis and had a validated method of health literacy assessment [55]. A prospective study on patients with head and neck cancer found a significant relation with education [56]. There was also a regional difference noted (people from outside the metropolitan areas having the longest TITD). These differences compared to ours could be explained by the different questionnaires for health literacy, the different health system accessibility, and possibly also different awareness among the patients about the health risks.

## 5. Strengths and Limitations

This study presents several notable strengths. Primarily, it investigates a relatively underexplored yet highly relevant aspect of lung cancer diagnostics: the influence of awareness of occupational exposure on diagnostic delays. To the best of our knowledge, it is the first one in Romania to address this gap in national research, and some of our findings have more general relevance. The use of a structured interview administered by an occupational physician enhances the accuracy and consistency of exposure assessment. The calculation of the sample size gave a fair statistical power to our results. The distribution of the histological types is rather similar with the ones reported in other tertiary-based Romanian studies, reflecting the general distribution by types of LC [57,58,59]. Moreover, the inclusion of a broad set of relevant variables such as smoking behavior, health literacy, family cancer history, and pulmonary comorbidities offers a well-rounded and in-depth analysis. The use of a validated health literacy instrument (HLS-EU-Q16) adapted to the Romanian population also adds methodological rigor and allows for comparability with other European studies. Furthermore, the study’s design, which includes both objective clinical data and patient reported variables, strengthens the validity of the findings.

However, several limitations should be acknowledged. The study used a convenience sample from two tertiary centers, which may limit generalizability to the broader LC population in Romania or other settings. The investigation of the time to diagnosis in tertiary medical settings offers the possibility to estimate the TITD, although it is not able to differentiate between patient and doctor driven delays. However, the full evaluation of a cancer patient (CT, bronchoscopy, biopsy, histology markers) is generally performed in hospitals, and, for the scope of our study (the TITD), patients recruited from these centers would provide the most accurate information. We are aware that the analysis should continue to also include the patients’ awareness about the risk, access and time to first appointment, the time to specialist referral and to the diagnostic procedures to define the most efficient interventions meant to overcome the stumbling blocks.

Recall bias is possible, particularly regarding self-reported occupational exposure and smoking history. The time from first exposure to diagnosis is long for solid occupational cancers, leading to a mean age at diagnosis of 55 years [60] or even higher [61]. This latency increases the chance of patients not mentioning their past exposure, particularly if not actively asked for. The cross-sectional design does not allow for causal inference between exposure and diagnostic delay. Additionally, although health literacy was assessed comprehensively, the instrument does not include items specific to occupational risk awareness, which may have limited its relevance to the primary research question.

Despite its limitations, this study offers important insights into how occupational exposure and individual patient characteristics influence the diagnostic timeline for lung cancer, underscoring the importance of involving occupational medicine in routine cancer management.

## 6. Clinical Implications

The results of the study emphasize the importance of knowing the patient’s occupational history and, more specifically, the length of exposure to pulmonary carcinogens. In order to reduce the TITD, there are several directions for improvement. First, is the communication of risk during the occupational exposure, the clarification of the long-term effects of exposure to carcinogens, and the need for follow-up after exposure cessation. Second, the post-exposure medical surveillance for which the good practice models existing in other countries should be generalized. Third, at health policy levels, occupational medicine should be better integrated in the general care to facilitate access for individuals exposed or that have been exposed to occupational carcinogens to lung screening, for early diagnosis. The actual disjunction is not in the best interest of the patient and even creates more financial burden for delayed treatment and for more extensive forms of cancer to treat. Fourth, these results also underline the need to apply a structured interview on these risk factors and to include, whenever possible, an occupational medicine physician in the clinical team in order to reduce the TITD [61]. The medical community should be more aware of these risks through their educational programs, interdisciplinary consultations should become more often in practice, and working in medical silos should be gradually abandoned.

## 7. Conclusions

Previous exposure to occupational hazards is important in assessing the individual risk for lung cancer. Awareness about the occupational risk in workers who are or have been exposed and in health providers could reduce the total interval time to diagnosis. Better occupational history and/or collaboration with an occupational physician would lead to better screening, assuring the therapeutic intervention in an earlier stage and increasing the survival and quality of life.

## Figures and Tables

**Figure 1 healthcare-14-00106-f001:**
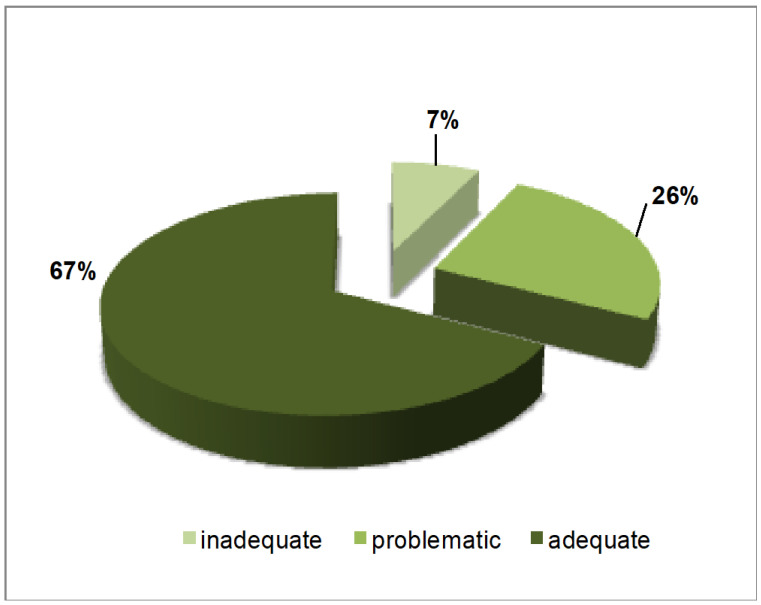
Distribution of level of health literacy in the total sample.

**Table 1 healthcare-14-00106-t001:** General characteristics and comparison of the study groups.

	Total	Cases	Controls	*p*
Age at dg (mean ± SD)	63.35 ± 10.22	63.63 ± 8.81	63.19 ± 10.96	0.71
Gender (no of Women, %)	44 (40%)	12 (31.57%)	32 (44.44%)	0.19
Level of education (no. of undergraduate, %)	81 (73.64%)	32 (84.21%)	49 (68.05%)	0.07
Smokers (no, %)	85 (77.27%)	30 (78.94%)	55 (76.39%)	0.76
Heavy smokers (no, %)	77 (70%)	29 (73.31%)	48 (66.67%)	0.29
No. of years of smoking	31.8 ± 20.22	33.24 ± 20.75	31.36 ± 19.95	0.31
No. of packs-years (mean ± SD)	38.05 ± 30.28	39.24 ± 29.07	37.42 ± 31.09	0.77
TIITD (months, mean ± SD)	3.41 ± 5.12	4.60 ± 5.88	2.77 ± 4.58	0.03
No. of patients diagnosed > 3.5 months (no, %)	29 (26.36%)	16 (55.17%)	13 (44.83%)	0.006
Family history of pulmonary cancer (no, %)	11 (32.50%)	4 (10.52%)	7 (9.72%)	0.89
Pulmonary diseases associated with high risk of LC (no, %)	36 (32.72%)	18 (47.37%)	18 (25%)	0.02
HLS-EU-Q16 total score (mean, SD)	13.25 ± 2.97	12.97 ± 2.71	13.40 ± 3.10	0.24
Q related to the easiness to find out where to obtain healthcare help	3.32 ± 0.59	3.26 ± 0.60	3.43 ± 0.57	0.14
Q related to the understanding of health warnings about smoking	3.2 ± 0.76	3.21 ± 0.66	3.19 ± 0.81	0.82
Q related to understanding doctors’ messages	3.2 ± 0.78	3.15 ± 0.71	3.22 ± 0.81	0.50
Q related to usage of the information the doctor gives in making decisions	3.26 ± 0.64	3.16 ± 0.68	3.32 ± 0.62	0.22

TITD = total diagnosis interval time; LC = lung cancer; HLS-EU-Q16 = European Health Literacy Survey Questionnaire, the version with 16 items; Q = question.

**Table 2 healthcare-14-00106-t002:** Results of the multivariate logistic regression analysis.

VAR	Odds Ratio	LCL	UCL	*p*
TITD cases/controls (controls as reference)	3.14	1.22	8.08	0.02
Age at diagnosis	0.96	0.92	1.01	0.13
Sex (women as reference)	1.33	0.45	3.96	0.61
Level of education	0.90	0.64	1.27	0.56
No of packs-years	0.99	0.98	1.02	0.81
Family history of cancer	0.25	0.03	2.17	0.21
HLS-EU-Q16 score	1.02	0.84	1.24	0.84
Pulmonary diseases associated with high risk of LC	1.02	0.38	2.71	0.97

## Data Availability

Data available on request due to restrictions.

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
