# Peer review of "Neglected Occupational Risk Factors—A Contributor to Diagnostic Delays in Lung Cancer"

_healthcare, 2026, doi:10.3390/healthcare14010106_

Round 1
Reviewer 1 Report
Comments and Suggestions for Authors
There is MUCH that need serious revision. Abstract line 32 needs clarification of "this" time. Line 39 needs +/-- some time to get an average had to be shorter. need range. Many typos or incorrect use of language throughout the paper. Line 91- by all cell type did this include carcinoids or only small cell and non-small cell cancers? Line 102 not clear with numbers given. Line 130---is there any accepted standard. lines 172/3 not clear 40 should be 1-4 months, since group less than 1-ie 30 . line 180-why 10 years-some carcinogens with much less exposure can cause disease. A major flaw of this study is no analysis by latency. Latency an important issue-and should have been analyzed. Line 211/12 important and needs fuller discussion at end. Lines 247/8- what are elementary and intermediate professions. Lines 283/4 often family members go into same trades so with this small group few family cases not really relevant and quality of history taking may have been poor. Line 325-what are causes of COPD if not smoking in these cases??? Line 337-COPD usually refers to both Chronic Bronchitis AND Emphysema-is COPD here used to mean CB. The issue in regard to health literacy is much less what the patient knows/ says/asks but the poor medical literacy of most doctors asking about occupational exposures . Conclusions-better histories would lead to better screening which would reduce late appearing ,less treatable cancers.
Comments on the Quality of English LanguageThe English needs significant work with poor words use or typos not caught. Someone more proficient in English needs to review.
Author Response
There is MUCH that need serious revision.
We are very grateful for your detailed observations. We have passed though all and made the appropriate clarifications and/or modifications.
- Abstract line 32 needs clarification of "this" time.
Thank you for your comment. We have changed “this time” with TITD
- Line 39 needs +/-- some time to get an average had to be shorter. need range.
Many typos or incorrect use of language throughout the paper.
Indeed, this was a typo mistake thanks for noticing it. We have changed it.
- Line 91- by all cell type did this include carcinoids or only small cell and non-small cell cancers?
We did not intended to exclude any form of histological cancer. We have specified this in the methods section by adding; “There were no exclusion criteria based on cellularity”
and in the results section by presenting the percentage of the histological types.
“Based on the histological classification, 44.94% were adenocarcinoma, 33.71% squamous cell carcinoma, 15.73% small cell lung carcinoma and 5.62% other forms”.
- Line 102 not clear with numbers given
Thank you for underlying the need for clarification. We have added the following lines in the methodology section:
Recent publications have enlarged the indication for screening from the initial threshold of 20 pack-year to only a longer than 20 years of smoking, even with less cigarettes smoked/day. Therefore, we have included in the group of high risk related to smoking both elements: either more than 20 pack years or more than 20 years of smoking.
- Line 130---is there any accepted standard.
As mentioned in the introduction, there are some general recommendations, but these recommendations should not be applicable for those at high risk, for whom screening is necessary.
The only national recommendation is to obtain the treatment plan set by the multidisciplinary oncologic commission in maximum 28 days from diagnosis. This recommendation does not fit in the total time to diagnosis analyzed in this article.
- lines 172/3 not clear 40 should be 1-4 months, since group less than 1-ie 30 .
Thank for this observation:
We have modified in the text. The graphical representation was removed to supplementary material, as recommended by another reviewer.
- line 180-why 10 years-some carcinogens with much less exposure can cause disease. A major flaw of this study is no analysis by latency. Latency an important issue-and should have been analyzed.
Thank you for raising this issue and giving us the opportunity for more detailed explanations.
We have included a paragraph explaining the choice of duration of exposure and not of the latency in the methods part. In the discussion section we have also .added a comment on this topic.
In the methods section:
There are currently different approaches for screening in occupational medicine; most of them recognize 10 years as minimum exposure [25,26,27] except for very high exposure that is also mentioned in some national recommendations [26]. However, in a large cohort highly exposed workers, 5 years of exposure was not sufficient to support the implementation of the low dose CT screening [28]. The duration of exposure time is a significant element in calculating the lifetime cumulative exposure. Therefore, we have chosen the 10 years of exposure to be a better estimate of the high risk category.
The latency for solid cancers is generally long, estimated to be more than 8-10 years [29,30]. An exposure time of 10 years would definitely include this latency and therefore will be a better indicator of high risk, as used in the screening recommendations described above. Even more, in the analysis of residents with LC (1042 exposed and 2364 controls, in Stockholm area), the sensitivity analysis based on the latency of exposure did not influenced the overall results [31], an argument in favor to our classification.
In the discussion section, namely in the limitation part we have added:
The time from first exposure to diagnosis is long for solid occupational cancers, leading to a mean age at diagnosis of 55 years [61] or even higher [62]. This latency increases the chance of patients not mentioning their past exposure, particularly if not actively asked for.
- Line 211/12 important and needs fuller discussion at end.
Thank you for raising this topic. This has included in the last part (clinical implications), which has been re-phrased as following:
The results of the study emphasize the importance of knowing the patient's occupational history and more specifically, the length of exposure to pulmonary carcinogens. In order to reduce the TITD, there are several directions for improvement. First, is the communication of risk during the occupational exposure, the clarification of the long term effects of exposure to carcinogens and the need of follow up after exposure cessation. Second, the post exposure medical surveillance for which the good practice models existing in other countries should be generalized. Third, at health policy levels, occupational medicine should be better integrated in the general care to facilitate access for individuals exposed or have been exposed to occupational carcinogens to lung screening, for early diagnosis. The actual disjunction, is not in the best interest of the patient and, even creates more financial burden for delayed treatment and for more extensive forms of cancer to treat. Forth, these results also underline the need to apply a structured interview on these risk factors and to include, whenever possible, an occupational medicine physician in the clinical team in order to reduce the TITD [62]{Citation}. The medical community should be more aware of these risks through their educational programs; the interdisciplinary consultations should become more often in practice, and working in medical silos should be gradually abandoned.
- Lines 247/8- what are elementary and intermediate professions.
Thank you for this observation. Classification of occupations are frequently used in occupational medicine and coding might be different. Therefore, for clarity, we have included the definitions from the reference. Following the description of the results from the Switzerland, study, we added:
This study counted in the category of elementary professions “workers in agriculture, industry and crafts with basic level of education” and in the intermediate category “administrative employees and service personnel and intermediate technical occupations which generally do not involve planning or supervisory powers and in the managerial categories”.
- Lines 283/4 often family members go into same trades so with this small group few family cases not really relevant and quality of history taking may have been poor.
We have extracted from the medical files the family history of cancer, which we think is relevant. In view to your comment, we have added to the discussion the following:
In interpreting this difference, we cannot exclude a possible bias coming from similar occupations (or occupational exposures) in the family members as we did not recorded the occupation of the family members. In order to clarify if this is a bias or not, further investigation, including genetic markers, focused on this topic is needed.
- Line 325-what are causes of COPD if not smoking in these cases???
In order to clarify this topic, we added the followings:
In non-smokers, the prevalence and risk factors to develop COPD have been recently reviewed [49]. The most frequent causes of COPD in non-smokers were biomass smoke exposure from cooking and heating, occupational exposures and outdoor air pollution [49] While these causes are relevant for COPD, they are also relevant for LC [50,51]. There is no clear relation between the airflow limitation and the risk for LC, but the extension of emphysema increases the risk [41,52]. This relation is even stronger when emphysema is associated with lung fibrosis [53]. By itself, the presence of the lung fibrosis denotes a high risk of LC [54].
- Line 337-COPD usually refers to both Chronic Bronchitis AND Emphysema-is COPD here used to mean CB.
Thank you for this remark
We agree that COPD would cover all elements (including emphysema) and therefore we have changed accordingly:
This is not a surprising result as all the occupational hazards to which these patients have been exposed could lead either to chronic bronchitis, to emphysema, or to lung fibrosis.
- The issue in regard to health literacy is much less what the patient knows/ says/asks but the poor medical literacy of most doctors asking about occupational exposures .
Thank you for this comment. The health literacy was included in the analysis because the lack of education and understanding of the risk factors leads to delay in diagnosis. It was namely used to asses a possible bias of the relation. In conjoint with this remark and the following one in the conclusion part we have underlined the need for awareness in the medical community.
- Conclusions-better histories would lead to better screening which would reduce late appearing ,less treatable cancers.
Thank for the synthesis of our results. We have included in the conclusions
Better occupational history and/or collaboration with an occupational physician, would led to a better screening, assuring the therapeutic intervention in an earlier stage and increasing the survival and quality of life.
Reviewer 2 Report
Comments and Suggestions for Authors
This an original and relevant study on diagnostic delay in lung cancer linked to occupational exposure. The topic addresses a documented gap in routine risk assessment. The dataset offers value. The analysis shows internal consistency. The manuscript fits the journal scope. The study design supports the main claim. The paper needs targeted revisions to reach publication standard. Some comments are listed below for authors consideration
- he manuscript state novelty but do not clearly position the study against recent European evidence. Several cited studies appear in later sections but not in the Introduction. Move key comparisons forward. Explicitly state how this study advances prior occupational lung cancer delay research rather than restating known underdiagnosis.
- The manuscript alternate between TITD and TDIT. This weakens clarity and risks editorial correction.
- Exposure classification transparency: Two occupational physicians classified exposure. The process lacks detail. Readers need clarity on disagreement handling.
- The manuscript rely on a convenience sample from two tertiary centers. This limitation appears late and understated.
- The manuscript report no association between health literacy and diagnostic delay. The discussion overextends interpretation.
- Smoking receives extensive discussion. The manuscript risks overstating independence from smoking.
- Correct typographical inconsistencies in spacing and hyphenation
Standardize p value formatting
• Replace vague phrasing such as “important factor” with specific outcomes
• Remove redundant citations already established earlier in the manuscript
• Shorten sections 4.3 and 4.4 by one third to improve focus - Occupational exposure adjudication lacks transparency
Location: Page 3, Lines 135–139
“two independent occupational physicians classified the patients according to their declared exposure”. Add a dedicated Methods subsection titled Exposure adjudication. State whether classifications were blinded. Report inter-rater agreement using Cohen’s kappa. Describe how disagreements were resolved. - Outcome terminology inconsistency
Pages 1–2, Abbreviations section
“total interval time to diagnosis (TITD)” and later “total diagnosis interval time (TDIT)”
Select one term. Apply it consistently across text, tables, figures, and abbreviations. - Convenience sampling weakens external validity
Page 10, Lines 377–380
“a convenience sample from a two tertiary centers”. Move this limitation to the opening sentence of the Limitations section. State referral bias risk and limits to generalizability explicitly. - Health literacy interpretation exceeds instrument scope
Page 9, Lines 349–356 “An apparently easy explanation of the delay… could be the difference in level of education”. Remove speculative framing. State directly that HLS-EU-Q16 lacks occupational risk awareness items and therefore limits inference on exposure-related delay. - Smoking interaction framed too strongly
Page 8, Lines 273–279
“in the multivariate analysis it did not influence the link”. Add one clarifying sentence stating absence of statistical association does not exclude biological synergy. Reference interaction literature already cited. - Exposure latency under-integrated into analysis
Page 7, Lines 233–239. “latency… can reach more than 30 years after the first exposure”
Explicitly link latency to diagnostic delay by stratifying or discussing age at exposure cessation versus diagnosis timing. - Figure 2 and Page 5, Lines 171–175
“The distribution according to the TDIT is presented in Figure 2.”
Move Figure 2 to supplementary material or reduce the Results narrative to avoid duplication. - Regression model justification incomplete
Page 6, Lines 206–208, Table 2. “Binary regression was calculated with covariates…”
Add a one-line rationale for covariate selection. State whether multicollinearity was assessed.
Author Response
This an original and relevant study on diagnostic delay in lung cancer linked to occupational exposure. The topic addresses a documented gap in routine risk assessment. The dataset offers value. The analysis shows internal consistency. The manuscript fits the journal scope. The study design supports the main claim. The paper needs targeted revisions to reach publication standard. Some comments are listed below for authors consideration
Thank you very much for the appreciation. We passed through all your comments and made the appropriate modifications.
- The manuscript state novelty but do not clearly position the study against recent European evidence. Several cited studies appear in later sections but not in the Introduction. Move key comparisons forward. Explicitly state how this study advances prior occupational lung cancer delay research rather than restating known under diagnosis.
We have moved these paragraphs in the introduction section:
The interval time to diagnosis and treatment is an important factor of survival, as the time for doubling of the tumor size varies between 70 and 260 days [10]. The non-adenomatous forms have a faster rate of growth, but the distinction is not always obvious before the histology result is available. Therefore, as a matter of precaution, the shorter TITD is considered the better.
The pathway from first symptoms to diagnosis and treatment in LC is influenced by a multitude of variables. It includes both the patient interval (time from symptoms to first visit) and the doctors’ interval (time from first visit to diagnosis) [11]. Both intervals are influenced by many actors of the medical and social system. Part of this delay is related to the access to the medical care, to the self-perception of the risk and of the personal symptoms management [12]. The occupational risk is generally underestimated by the workers [13]. Openness and emotional stability seem to be associated with a reduction in risk perception, while conscientiousness and friendliness with an increase [14].
- The manuscript alternate between TITD and TDIT. This weakens clarity and risks editorial correction.
Thank you for this remark, we have made the necessary changes.
- Exposure classification transparency: Two occupational physicians classified exposure. The process lacks detail. Readers need clarity on disagreement handling. “two independent occupational physicians classified the patients according to their declared exposure”. Add a dedicated Methods subsection titled Exposure adjudication. State whether classifications were blinded. Report inter-rater agreement using Cohen’s kappa. Describe how disagreements were resolved.
Thank you for your remark. We have added in the methodology section this description and in the results section the level of agreement.
Classification was blinded for both evaluators. Each expert classified the exposure based on hazards (namely the ones included by IARC in the class I category), occupation and occupational domain. The inter-rater agreement was calculated with MedCalc statistical software [21]. When disagreement was met, a consensus was reached using as references for occupations/work process listed in the existing international data bases or published in scientific literature [22,23,24].
In the results part, we added the following:
The inter-rate agreement between the two evaluators was very high (kappa= 0.92848, standard error =0.03512 and 95%CI: 0.85965 - 0.99731). For the 4 cases which were differently classified, a consensus was reached by discussing the initial opinions and confronting them with the most literature references for that particular patient.
- The manuscript rely on a convenience sample from two tertiary centers. This limitation appears late and understated.
We have modified in the methodology section, as following:
Although a convenience sample is not representative for the whole patients’ population, in order to reduce possible biases, we included patients from the largest hospital in Bucharest, fully dedicated to lung diseases (Marius Nasta Institute of Pulmonology) and a large general, multidisciplinary hospital. Both hospitals had pulmonology and oncology departments and the patients diagnosed with cancer were also receiving the adjuvant oncological treatment and followed periodically.
In the limitation part we added the phrases:
The investigation of the time to diagnosis in tertiary medical settings offers the possibility to estimate the TITD, although it is not able to differentiate between patient and doctor driven delays. However, the full evaluation of a cancer patient (CT, bronchoscopy, biopsy, histology markers ) is generally performed in hospitals, and, for the scope of our study (the TITD) patients recruited from these centers would provide the most accurate information. We are aware that the analysis should continue to include also the patients’ awareness about the risk, access and time to first appointment, the time to specialist referral and to the diagnostic procedures, to define the most efficient interventions meant to overcome the stumbling blocks.
.
- The manuscript report no association between health literacy and diagnostic delay. The discussion overextends interpretation.
Thank you for this remark; it has been indeed a surprise for us as well. The health literacy was included in the analysis because the lack of education and understanding of the risk factors leads to delay in diagnosis. It was namely used to exclude a possible bias of the relation.
- Smoking receives extensive discussion. The manuscript risks overstating independence from smoking.
We had no intention to underestimate the risk of smoking. Therefore, in accordance with your remark no 12 we added at the end of the paragraph dedicated to smoking, the followings:
In our sample, smoking did not influenced the link between LC and occupational carcinogens. This result just underlines the need to proceed to a definitive diagnosis whenever symptoms suggestive for LC are present. The lack of statistical association does not exclude the biological relation and the additive or synergic effects of smoking with the occupational hazards, supported by extensive medical data [44]and do not minimize the role of smoking itself in the risk assessment.
- Correct typographical inconsistencies in spacing and hyphenation
Standardize p value formatting
Replace vague phrasing such as “important factor” with specific outcomes
• Remove redundant citations already established earlier in the manuscript
• Shorten sections 4.3 and 4.4 by one third to improve focus
We have shorten 4.3. but for the 4.4 we had different comments from another reviewer and because it was anyway very long, we did not shorten it so much.
This also removed several references.
- Occupational exposure adjudication lacks transparency
Location: Page 3, Lines 135–139
Please consider the modifications mentioned above.
- Outcome terminology inconsistency
Pages 1–2, Abbreviations section
“total interval time to diagnosis (TITD)” and later “total diagnosis interval time (TDIT)”
Select one term. Apply it consistently across text, tables, figures, and abbreviations.
Thank you for this observation. We have carefully revised this terminology and made the necessary changes
- Convenience sampling weakens external validity
Page 10, Lines 377–380
“a convenience sample from a two tertiary centers”. Move this limitation to the opening sentence of the Limitations section. State referral bias risk and limits to generalizability explicitly.
As recommended we have explained the rationality of the selection process in the methods part and mentioned this limitation for the identification of a more specific cause (patient or doctor driven) in the assessment of the delay.
- Health literacy interpretation exceeds instrument scope
Page 9, Lines 349–356 “An apparently easy explanation of the delay… could be the difference in level of education”. Remove speculative framing. State directly that HLS-EU-Q16 lacks occupational risk awareness items and therefore limits inference on exposure-related delay.
Thank you for the comment. We agree that a speculative framing is not appropriate and removed it.
- Smoking interaction framed too strongly
Page 8, Lines 273–279
“in the multivariate analysis it did not influence the link”. Add one clarifying sentence stating absence of statistical association does not exclude biological synergy. Reference interaction literature already cited.
Please refer to the answer to your comment no 6.
- Exposure latency under-integrated into analysis
Page 7, Lines 233–239. “latency… can reach more than 30 years after the first exposure”
Explicitly link latency to diagnostic delay by stratifying or discussing age at exposure cessation versus diagnosis timing.
Thank you for this comment: We have included in the methods part the distinction between exposure time and latency and in the discussion section the explanation that this might be a factor influencing the delay.
Paragraphs added in the methods part:
There are currently different approaches for screening in occupational medicine; most of them recognize 10 years as minimum exposure [25,26,27] except for very high exposure that is also mentioned in some national recommendations [26]. However, in a large cohort highly exposed workers, 5 years of exposure was not sufficient to support the implementation of the low dose CT screening [28]. The duration of exposure time is a significant element in calculating the lifetime cumulative exposure. Therefore, we have chosen the 10 years of exposure to be a better estimate of the high risk category.
The latency for solid cancers is generally long, estimated to be more than 8-10 years [29,30]. An exposure time of 10 years would definitely include this latency and therefore will be a better indicator of high risk, as used in the screening recommendations described above. Even more, in the analysis of residents with LC (1042 exposed and 2364 controls, in Stockholm area), the sensitivity analysis based on the latency of exposure did not influenced the overall results [31], an argument in favor to our classification.
In the discussion section, namely in the limitation part:
The time from first exposure to diagnosis is long for solid occupational cancers, leading to a mean age at diagnosis of 55 years [61] or even higher [62]. This latency increases the chance of patients not mentioning their past exposure, particularly if not actively asked for.
- Figure 2 and Page 5, Lines 171–175
“The distribution according to the TDIT is presented in Figure 2.”
Move Figure 2 to supplementary material or reduce the Results narrative to avoid duplication.
As suggested we moved the Fig 2 to the supplementary material.
- Regression model justification incomplete
Page 6, Lines 206–208, Table 2. “Binary regression was calculated with covariates…”
Add a one-line rationale for covariate selection. State whether multicollinearity was assessed.
Thank you for raising this issue. We added in the methods section the following lines:
Binary regression was calculated for the association between high occupational risk and delayed TITD. Further on, we verified the influence of other variables on this correlation. Beside the basic demographic data (age, gender) we included other well defined risk factors for LC (number of packs-years, pulmonary diseases associated with LC, and family history of cancer). The score of health literacy was used in order to exclude a possible bias derived from possible delays generated by the patient lack of understanding or risk factors or of doctors’ recommendations. Before proceeding to the multivariate regression, collinearity of the variables has been checked.

Round 2
Reviewer 1 Report
Comments and Suggestions for Authors
revisions as submitted have markedly improved the paper.
Comments on the Quality of English LanguageThe English needs significant work with poor words use or typos not caught. Someone more proficient in English needs to review.
Reviewer 2 Report
Comments and Suggestions for Authors
Thank you. No further comments